# A Holistic Approach to Hard-to-Treat Cancers: The Future of Immunotherapy for Glioblastoma, Triple Negative Breast Cancer, and Advanced Prostate Cancer

**DOI:** 10.3390/biomedicines11082100

**Published:** 2023-07-25

**Authors:** Carles Puig-Saenz, Joshua R. D. Pearson, Jubini E. Thomas, Stéphanie E. B. McArdle

**Affiliations:** 1The John van Geest Cancer Research Centre, School of Science and Technology, Nottingham Trent University, College Drive, Clifton, Nottingham NG11 8NS, UK; carles.puigsaenz2016@my.ntu.ac.uk (C.P.-S.); jrdpearson@hotmail.co.uk (J.R.D.P.); jubini.thomas2020@my.ntu.ac.uk (J.E.T.); 2Centre for Systems Health and Integrated Metabolic Research, School of Science and Technology, Nottingham Trent University, College Drive, Clifton, Nottingham NG11 8NS, UK

**Keywords:** glioblastoma, triple negative breast cancer, advanced prostate cancer, microbiota, stress, exercise, immunotherapy, vaccine

## Abstract

Immunotherapy represents an attractive avenue for cancer therapy due to its tumour specificity and relatively low frequency of adverse effects compared to other treatment modalities. Despite many advances being made in the field of cancer immunotherapy, very few immunotherapeutic treatments have been approved for difficult-to-treat solid tumours such as triple negative breast cancer (TNBC), glioblastoma multiforme (GBM), and advanced prostate cancer (PCa). The anatomical location of some of these cancers may also make them more difficult to treat. Many trials focus solely on immunotherapy and have failed to consider or manipulate, prior to the immunotherapeutic intervention, important factors such as the microbiota, which itself is directly linked to lifestyle factors, diet, stress, social support, exercise, sleep, and oral hygiene. This review summarises the most recent treatments for hard-to-treat cancers whilst factoring in the less conventional interventions which could tilt the balance of treatment in favour of success for these malignancies.

## 1. Introduction

Immunotherapy, such as the use of checkpoint inhibitors, oncolytic virus, bispecific antibodies, and adoptive cell transfer, has revolutionised the treatment of many cancers. Today, many of these approaches have been FDA approved for the treatment of several cancers [1]. Talimogene laherparepvec (T-VEC) is an oncolytic virus, FDA approved in 2015, for the treatment of advanced melanoma (stage IIIB-IV) [2] and Mosunetuzumab-axgb is a bispecific antibody approved for the treatment of adults with relapsed or refractory follicular lymphoma; in addition, three other bispecific antibodies have recently been approved for the treatment of acute lymphoblastic leukaemia (Blincyto), for non-small cell lung cancer (Rybrevant), and for uveal melanoma (Kimmtrak). In terms of cellular therapy, FDA-approved treatment using chimeric antigen receptor T (CAR-T) has been limited to B cell malignancies expressing CD19 and, while these harbour high potential for solid tumours, they also have significant toxicity, including severe cytokine release syndrome (CRS) and substantial neurotoxicity. Additionally, PROVENGE is the only FDA-approved immunotherapy for the treatment of advanced prostate cancer (PCa) without DNA mismatch repair deficiency (dMMR); however, while the therapy is well tolerated, it remains expensive and limited in its efficacy. A vast number of additional approaches (neoantigens with adjuvants, dendritic cell (DC) vaccines, peptide/mRNA vaccines…) are being investigated, including any combinations of the aforementioned, FDA-approved treatments.

The immune system is known to be strongly affected by both intrinsic (age, sex, and genetics) and extrinsic (environmental and lifestyle) factors. While one cannot change an individual’s intrinsic factors, external factors, especially those linked to lifestyle, can be manipulated. Yet very few studies have attempted to combine “conventional approaches” with any of these. In this review, we summarise the difficulties faced when treating glioblastoma (GBM), triple negative breast cancer (TNBC), and PCa, and how environmental and lifestyle factors represent emerging parameters that strongly influence the progression of these diseases. We also discuss how some of these parameters have the potential to be added prior to and along with conventional therapy to increase overall survival and/or quality of life for the patients.

## 2. Glioblastoma (GBM)

GBM is the most frequently occurring primary brain tumour with an incidence rate of ~3 in 100,000. It affects both children and adults, although it is primarily a disease associated with increased age, with the average age of diagnosis being ~65 years. GBM carries a poor prognosis and is nearly always fatal, with only around 3–5% of patients surviving for a period of five years or more [3,4]. Current therapy involves surgical resection (where possible) followed by concomitant radiotherapy and temozolomide chemotherapy. Despite aggressive multimodal therapy, nearly all tumours recur close to the site of resection. Complete surgical resection is almost impossible due to the highly infiltrative nature of GBM [5]. In addition, not all tumours are responsive to temozolomide chemotherapy, and some patients’ tumours may express the enzyme O6-methylguanine-DNA-methyltransferase (MGMT). MGMT repairs the DNA damage induced by temozolomide, making patients whose tumours express MGMT resistant [6]. Very few new therapies have been approved for GBM in recent years and, due to the poor prognosis associated with GBM, new therapeutic interventions are desperately required. Immunotherapy represents an attractive therapy option due to its tumour specificity and the ability of activated immune cells to access the brain and target intracranial tumours. Numerous immunotherapeutic approaches are being trialled in the GBM setting (Table 1); however, currently there are no approved immunotherapies for GBM. While numerous successes have been seen in other cancers, as yet no immunotherapy has been approved for use in GBM, with several therapies failing to show efficacy during phase 3 testing (e.g., Rindopepimut) [7]. This could be due to a highly immunosuppressive tissue microenvironment, the inability of large molecules to cross the blood–brain barrier and penetrate tumours, and the low mutation rates compared to other tumours. T-cell dysfunction is also frequently seen in the GBM setting with cells often expressing exhaustion markers and having an altered metabolism [8,9,10]. Standard therapies used to manage GBM are also known to dampen the immune response, with one such example being corticosteroids used to treat GBM-associated oedema. 

Research has begun to highlight the importance of several lifestyle factors when utilising the immune system to fight GBM. The gut microbiome has been identified as an important component of the immune system and a predictor of response to immunotherapy in the cancer setting. Indeed, the gut microbiome has been shown to be a predictor of response to anti-PD-1 immune checkpoint blockade in the murine GBM setting, with the presence of *Bacteroides cellulosilyticus* being linked with response to anti-PD-1 immune checkpoint therapy [65]. 

The alteration of patient diets has been examined as a therapeutic intervention in the cancer setting [66]. Several studies have looked at utilising a ketogenic diet to treat GBM with the aim of reducing carbohydrate intake and therefore starving the tumour cells of glucose. Furthermore, it seems as if a ketogenic diet can enhance immunity in a murine GBM model. Mice harbouring intracranial GL261-Luc2 tumours were given a ketogenic diet and it was found that these mice had a significant reduction in immune inhibitory receptors such as PD-1 and CTLA-4 among their tumour-infiltrating lymphocytes [67]. This information points to a potential combinatorial role for the ketogenic diet with active immunotherapy. 

Exercise also appears to have a potential role in the therapy of GBM. A case study reported that exercise improved the quality of life for a patient undergoing radiation treatment and she displayed signs of increased physical fitness such as muscle strength, balance, and aerobic capacity [68]. These impacts on the quality of life can also improve the psychological status of the patient, which may also have a knock-on effect on the anti-tumour immune response. Furthermore, irisin, a myokine associated with exercise, was shown to have anti-GBM properties in vitro, leading to cell cycle arrest in a panel of three cell lines. In vivo injection of irisin into a U-87 MG tumour led to a reduction in tumour size compared to untreated tumours and, even more impressively, mice that exercised on a running wheel also had reduced tumour growth compared to controls [69]. 

Several clinical trials investigate the effects of exercise, diet, and the microbiome on GBM (Table 2). Many of these holistic approaches have been examined as potential therapies on their own; however, several of these therapies could be used with active immunotherapy to boost the anti-tumour immune response and improve patient outcomes. 

## 3. Triple Negative Breast Cancer (TNBC)

TNBC is a hard-to-treat type of breast cancer characterised by the lack of oestrogen receptor alpha (ERa), progesterone (PR), and HER2 receptors. Around 15% of all breast cancers fall under the category of triple negative [85], are more prevalent in black women, premenopausal women, women under 40 years of age, and women carrying *BRCA1* mutations [86], and are extremely rare in men [87]. According to the Surveillance, Epidemiology, and End Result Program (SEER) database, TNBC bears an overall 5-year survival rate of 77% which drops down to 12% when the disease is at the metastatic stage [88]. In non-metastatic TNBC, between 30 and 40% of cases will result in relapse often leading to metastasis [89,90].

Compared to other types of breast cancer, the treatment options for TNBC are limited due to the lack of targetable receptors. ERa, PR, and HER2 are well known to play critical roles in the tumorigenesis of breast cancers, acting as therapeutic targets for a large proportion of patients [91,92]. Conventional treatments for TNBC include breast-conserving surgery and mastectomy, usually followed by radiotherapy and/or chemotherapy. Common systemic agents include anthracyclines, platinum-based drugs, and taxanes [93]. Interestingly, despite the overall aggressiveness of TNBC, a significant proportion of patients achieve a pathologic complete response following neoadjuvant chemotherapy [94]; however, incomplete responses are associated with high risk of recurrence [95]. 

Due to the high heterogeneity of TNBC tumours, finding a common targeted therapy for all TNBC becomes challenging. Although none have yet been approved, some targeted therapies currently under investigation include poly (ADP-ribose) polymerase (PARP) inhibitors, which induce cell death in cells with *BRCA* mutations; androgen receptor antagonists, which stunt the growth of TNBC subtypes expressing androgen receptor; antiangiogenic agents such as vascular endothelial growth factor receptor (VEGFR) inhibitors, which block the recruitment of new blood vessels towards the tumour; and epigenetic regulators such as DNA methyltransferase and histone deacetylase inhibitors, which have shown the capacity to induce the expression of oestrogen receptors, sensitising tumours to hormone therapy [96]. In addition to these therapies, many vaccines are being trialled for the treatment of TNBC (Table 3).

Disruptions in the balanced diversity of the microbiota, referred to as microbiome dysbiosis, is known to contribute to several health disorders [113,114]. Advances in meta-omics research technologies are facilitating our understanding of how this phenomenon can lead to other less understood disorders, including cancer [115]. Additionally, attention is increasingly being paid to the modulating effects of the microbiome on treatments such as cancer immunotherapies [116]. Although not yet elucidated, increasing evidence suggests that dysbiosis may contribute to the pathogenesis of breast cancer in various ways [117]. Overall, the consensus is that cancers are associated with a reduced diversity in gut microbiota [118], with studies suggesting that breast cancer is not an exception [119,120,121]. 

Oestrogen metabolism is a potential mechanism by which the gut microbiome can influence breast cancer pathogenesis. Whether endogenous or exogenous, oestrogen is a known risk factor for breast cancer, particularly in postmenopausal women [122]. The gut microbiome seems to play a role in oestrogen-driven breast cancers by enzymatically deconjugating oestrogen, therefore forcing it back into circulation and increasing systemic levels [123]. In TNBC, however, the negative impact of microbiota-mediated oestrogen abundance does not seem to occur, likely due to the lack of ERa expression. TNBCs have alternative oestrogen signalling pathways which render them responsive to circulating hormones [124]. Interestingly, studies suggest that receptors such as ERb, G protein-coupled oestrogen receptor 1, and oestrogen-related receptors have anti-cancer effects and that, when these are expressed in TNBCs, prognosis seems to be better [125]. 

Moreover, the microbiome does not extend exclusively to the gastrointestinal tract; in fact, different parts of the human body have been found to host different populations of microbes, and these can vary among and within individuals due to factors such as diet, lifestyle, usage of antibiotics, and even social interactions [126]. It is therefore not surprising that there exists a breast tissue-specific microbiome [127]. Studies such as that of Tzeng et al. have found that the most abundant phylum of bacteria in both healthy and cancerous breast tissue is Proteobacteria, while TNBC tissue was composed of bacteria from the genera *Azomonas*, *Alkanindiges*, *Caulobacter*, *Proteus*, *Brevibacillus*, *Kocurla*, and *Parasediminibacterium* [128]. However, previous studies have identified different genera in TNBC tissues [128,129,130,131], highlighting the need for further research and refinement of methodologies to study the microbiome.

Physical exercise has long been known to offer a wide range of health benefits [132]. Among these benefits, it is said to act as an “immune system adjuvant” which improves the recirculation and activity of certain immune components [133]. Preclinical evidence suggests that lifestyle can have a positive effect on the immune system when it comes to fighting cancer. Hojman et al. demonstrated a reduction in tumour growth in mice with access to voluntary wheel running, highlighting an increase in tumour immune recognition by macrophages, NK and T cells, but a decreased recognition in mice fed high-fat diets [134]. In a model of TNBC, Wennerberg et al. found a reduction in tumour-induced myeloid-derived suppressor cell (MDSC) recruitment, as well as an increase in NK and CD8^+^ T cell activation in the exercise treatment group, including an improvement of response to PD-1 inhibition [135]. Obesity has a tumorigenic effect in TNBC. For example, it alters the immune response by reprogramming mammary adipose tissue macrophages to a pro-inflammatory metabolically activated phenotype [136]. It also contributes to metabolic dysregulation, with evidence suggesting that exercise can reduce tumour growth by means of metabolic—mitochondrial and macronutrient—regulation [137]. 

Clinical studies suggest that an improvement in overall and disease-free survival is observed following moderate physical exercise upon diagnosis [138,139], with obesity playing a negative role in the outcome of all subtypes of breast cancer [140]. Exercise may exert these effects in different ways. For example, decreasing kynurenine pathway metabolites [141]; this pathway is known to be dysregulated in TNBC, contributing to the inhibition of anti-tumour responses [142]. It may also have a positive impact on inflammatory cytokines [143] known to play a role in TNBC [144]. As the evidence mounts, it seems sensible to use the benefits of maintaining a healthy lifestyle in order to help prevent—and possibly be considered before and during immunotherapeutic interventions against—cancer. The World Cancer Research Fund and the American Institute for Cancer Research have published recommendations for cancer prevention, which include guidelines for physical activity as well as for healthy diets and weight [145]. Although further research is warranted, studies suggest that lifestyle changes have the potential to improve treatment response and risk of relapse [146], and clinical trials are being conducted to assess the effect of exercise, diet, and the microbiome in TNBC patients (Table 4).

## 4. Advanced Prostate Cancer (PCa)

PCa is one of the most diagnosed fatal malignancies among men worldwide [151]. Compared to other common cancers, the aetiology of PCa remains unknown. Advanced age, positive family history, prostate inflammation, obesity, lack of exercise, ethnicity, and persistent elevated levels of testosterone are some of the risk factors known for PCa [152]. 

Localised prostate cancer is primarily managed through active surveillance, radical prostatectomy, external radiotherapy, and brachytherapy [152]. However, there is evidence of biochemical recurrence of malignancy observed within ten years of initial treatment in approximately 30–50% of patients who received radiotherapy or 20–40% of patients who underwent prostatectomy [151]. The advanced stage of prostate cancer is typically treated with androgen deprivation therapy (ADT), which is effective in controlling cancer growth. However, most patients eventually progress to metastatic castration-resistant prostate cancer (mCRPC). The loss of testosterone resulting from ADT is often associated with intense side effects, including mood swings, erectile dysfunction, and loss of bone density. Other approved therapies for PCa, such as radium-223 and taxane chemotherapy, have shown limited improvement in overall survival for patients, as the cancer continues to progress [153]. 

Sipuleucel-T is the only FDA-approved cellular immunotherapy for PCa. This approach has shown to increase PCa patients’ overall survival by 4.1 months. This vaccine is prepared by collecting the patient’s peripheral blood mononuclear cells (PBMCs) through leukapheresis. The collected cells are then incubated ex vivo with PA2024, which is a recombinant fusion protein combining prostatic acid phosphatase (PAP) and granulocyte macrophage colony-stimulating factor (GM-CSF). Finally, the engineered product is reinfused back into the patient. The PAP antigen is specific to prostate tissue and is expressed in most prostate adenocarcinomas. However, the high cost of this treatment limits its widespread availability [154]. 

Immune checkpoint inhibitor treatments such as anti-PD1 have shown significant clinical benefit for PCa patients whose tumours harbour DNA mismatch repair deficiency (dMMR). However, these only account for 3–5% of all castration-resistant prostate cancer, and only have modest activity in unselected men with metastatic prostate cancer. It is highly likely that the limited clinical response of immunotherapy in PCa is due to the immunosuppressive tumour microenvironment (TME) associated with it. This environment is characterised by the presence of immunosuppressive cells such as tumour-associated macrophages, MDSCs, and regulatory T cells. Additionally, adenosine produced via PAP and transforming growth factor-β (TGF-β) act as potent immunosuppressive molecules [153]. Interestingly, among genitourinary malignancies, PCa exhibits a distinct TME profile. PCa-associated tumour intrinsic factors such as decreased MHC class I expression, low tumour-associated antigen expression, loss of tumour suppressor protein PTEN, dysfunctional signalling of type I interferons, and mutations in the DNA damage repair genes *BRCA1* and *BRCA2* contribute towards the evolution of immunologically cold PCa TME [155]. Furthermore, PCa biopsy samples have shown the presence of tumour-infiltrating lymphocytes (TILs) that are biased toward T-regulatory (Treg) and T-helper 17 (Th17) phenotypes, which suppress autoreactive T cells and anti-tumour immune responses [154]. Several immunotherapy trials for PCa are underway or have already been completed (Table 5).

Certain host factors such as composition of the gut microbiota may also facilitate PCa progression and impact response to chemotherapy and immunotherapy [154,162]. Occurrence of certain microorganisms such as *Cutibacterium* in human prostate can cause immunosuppression and prostatitis by stimulating the infiltration of CD4+FoxP3+ cells (Treg) and Th17 cells [162]. The composition of the gut microbiota also plays a significant role in the response elicited by ADT with its immunostimulatory or immunosuppressive and direct ADT subversion. Depletion of immunostimulatory gut microbiota by orally administered broad spectrum antibiotics in mouse models have been shown to diminish the efficacy of ADT [163]. 

The decreased levels of androgen due to ADT in PCa patients have been suggested to contribute to a reduction in α and β-diversity in gut microbiota, leading to the development of dysbiosis [163,164,165]. A study involving sequential faecal and blood samples collected from 23 PCa patients showed a significant difference in the abundance and composition of microbiota, including increased levels of Proteobacteria, *Pseudomonas*, and Gammaproteobacteria, after ADT compared to before ADT [164]. Certain intestinal bacteria have the ability to degrade ADT-relevant drugs, thereby reducing the effectiveness of the therapy. Specific gut microbiota can act as androgen-producing bacteria by converting androgen precursors into active androgen. The abundance of such microbiota has been observed in castrated mice as well as in patients with castration-resistant prostate cancer (CRPC). Interestingly, a significant reduction in circulating testosterone levels has been observed in castrated mice when their gut microbiota is depleted [163]. 

The abundance of gut microbiota that can interfere with the clinical responses to ADT ultimately leads to the development of CRPC [165]. In a study conducted by Liu and Jiang, the faecal microbiota of 21 patients who received ADT was profiled, revealing compositional differences in gut microbiota between hormone-sensitive prostate cancer and CRPC [165]. CRPC was found to have a significant increase in the abundance of fourteen phylotypes of microbial flora, including *Phascolarctobacterium* and *Ruminococcus*. Additionally, bacterial gene pathways involved in terpenoid/polyketide metabolism and ether lipid metabolism were notably activated in CRPC. Similarly, another study by Che et al. on faecal microbiota demonstrated significant differences in the abundance of bacteria between prostate cancer patients and healthy individuals, with metabolic pathways associated with folic acid and arginine being affected [166]. Folic acid is crucial for nucleotide synthesis and DNA methylation, and its deficiency can lead to DNA instability and increased mutation rates. Moreover, folic acid-producing microflora were found to be less abundant in PCa patients compared to non-cancer patients, suggesting that natural sources of folic acid may offer protection against prostate cancer [166]. 

Another contributing factor to intestinal dysbiosis is lifestyle, including factors such as diet and obesity, which are often associated with an increase in circulating levels of pro-inflammatory bacterial lipopolysaccharide (LPS), leading to the development of prostate cancer [162,163]. In mouse models, the accumulation of LPS has been shown to activate local inflammation and promote prostate tumour growth [162]. A diet high in saturated fat can also promote the progression of prostate cancer by increasing circulating levels of androgens and causing DNA damage in cells through elevated oxidative stress [167]. Additionally, a Western-style high-fat diet, which often leads to obesity, can induce chronic inflammation, and contribute to the development of prostate cancer by upregulating inflammatory cytokines such as IL-6 [163]. Clinical trials are looking at the effects of exercise, diet, and the microbiome on PCa (Table 6).

In addition to lifestyle factors, antibiotic exposure can also contribute to gut dysbiosis. Research conducted by Zhong et al. demonstrated that antibiotic exposure leads to an enrichment of gut Proteobacteria, increased gut permeability, and elevated levels of intra-tumoral lipopolysaccharide (LPS), which promote the development of prostate cancer through the NF-κB-IL6-STAT3 axis in mice [178]. 

Another interesting factor to consider is the signalling of β-adrenergic receptors (β-AR), which plays a vital role in the progression and metastasis of many cancers, including PCa [179]. Higher expression of β2-AR has been observed in carcinoma compared to normal prostate tissues, as observed in tissue microarray studies using immunohistochemistry [180]. In line with this finding, Zang et al. have uncovered a significant role for β2-AR signalling in regulating the activity of the Shh pathway in PCa tumorigenesis using xenograft models [180]. Moreover, the administration of propranolol, a nonselective β-AR blocker, has demonstrated anti-cancer effects in cancer cell lines and animal models [179]. 

A patient cohort study utilising the Taiwan National Health Insurance Research Database, covering the period from January 2000 to December 2011, examined the usage of propranolol in various cancers, including PCa. The study concluded that propranolol can reduce the risk of cancers, with the most substantial protective effect observed when propranolol usage exceeded 1000 days [179]. To understand the efficacy of β-blockers in PCa patient mortality, Grytli et al. conducted a study involving 3561 PCa patients, out of which 1115 patients used β-blockers before and after diagnosis [181]. The study found a reduction in cancer-specific mortality among high-risk or metastatic PCa patients who used β-blockers. 

Considering the potential therapeutic options suggested by targeting gut microbiota dysbiosis [164,165] and β2-adrenergic modulation [180], a combination of these approaches with other available therapeutic strategies could potentially benefit the management of PCa.

## 5. Chronic Stress and Cancer

Acute stress is a beneficial neuroendocrine response to external or internal stressor events which in turn activates our fight-or-flight response [182]. On the other hand, chronic stress is known to have detrimental effects on various aspects of physiology, with increasing research suggesting that it could lead to cancer progression [183]. The stress response begins when the amygdala perceives danger stimuli and relays this information to the hypothalamus which, in turn, promotes the release of catecholamines from the adrenal glands—a neuroendocrine component known as the sympathetic–adreno–medullar axis (Figure 1). In the TME, adrenaline and noradrenaline can have an immunosuppressive effect during chronic stress, for example, by increasing MDSC frequency [184]. In addition, stress-induced dopamine has recently been involved in tumour progression via activation of hypoxia-inducible factor-1α [185]. The hypothalamus also produces corticotropin-releasing factor, which acts upon the pituitary gland to promote the release of adrenocorticotropic hormone; this hormone then travels to the adrenal cortex to stimulate the synthesis and secretion of corticosteroids in what is called the hypothalamus–pituitary–adrenal axis. Corticosteroids are widely known for their immunosuppressive effects, affecting key effectors of anti-tumour immunity such as dendritic cells and T cells [186] (Figure 1).

It is not uncommon for patients living with cancer and other life-threatening diseases to feel stressed, anxious, and/or depressed. Given that these negative mental health states can worsen cancer prognosis, it seems appropriate to pay more attention to patient wellbeing; for example, offering patients resilience-enhancing interventions during key stages of their disease could accelerate their mental health recovery [187]. Moreover, research points to various links between diet and depression, for example, the consumption of ultra-processed foods [188] as well as diets rich in proline; with regards to the latter, Mayneris-Perxachs et al. recently found that a healthy gut microbiome is associated with lower plasma proline levels and lower depression scores, highlighting the importance of maintaining a balanced diet [189]. In addition, Valles-Colomer et al. found a positive association between the presence of *Dialister* and *Coprococcus* species and quality of life, with these species being sparse in depressed individuals [190]. Given the interplay between mental health, the immune system, diet, the microbiome, and cancer (Figure 1), it seems sensible, therefore, to approach the mental wellbeing of patients from a holistic point of view.

## 6. Conclusions and Future Directions for Immunotherapy

Our immune system needs to be able to respond appropriately to external and internal environmental changes caused by factors such as physical or psychological stress, nutrient availability, the microbiota, temperature, pathogens, and malignancies. To achieve this, immune cells are equipped with a plethora of mechanisms designed to recognise disruptions in homeostasis and to respond to these deviations. These in turn will be influenced by an individual’s genetic makeup and their history of antigen experience. We believe that antigen-specific immunotherapy, which aims at stimulating immune cells to target tumours, will be more successful if applied at a time when cancer cells are few within the body, helping to prevent relapse rather than to treat large tumours. Importantly, we believe that immunotherapy can be tilted towards a positive outcome if applied at a time when the patient’s entire wellbeing has first been taken into consideration and interventions have been taken to improve their mental health, gut microbiota, and approach to exercise prior to receiving therapy as well as during immunotherapeutic interventions. 

In a time of “precision medicine” where most scientists believe that the treatment needs to be tailored to each individual patient, we would like to put forward the idea that preparing patients physically, psychologically, and microbiologically will improve the potency of any immunotherapy.

## Figures and Tables

**Figure 1 biomedicines-11-02100-f001:**
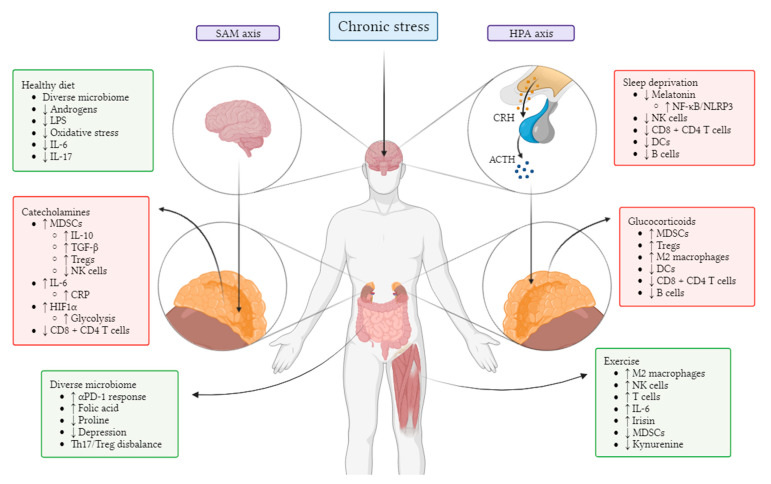
Mechanisms of immune modulation by physical and psychological factors. Abbreviations: SAM, sympathetic–adreno–medullar; HPA, hypothalamic–pituitary–adrenal; DCs, dendritic cells; MDSCs, myeloid-derived suppressor cells; Tregs, regulatory T cells; HIF1α, hypoxia-inducible factor-1α; CRP, C-reactive protein; TGF-β, transforming growth factor-β; LPS, lipopolysaccharide.

**Table 1 biomedicines-11-02100-t001:** Immunotherapy trials for GBM.

ClinicalTrials.gov ID	Vaccine	Phase	Stage	Reference
NCT02049489	ICT-121	I	Completed	[11]
NCT00323115	aDCs ^1^ + RT ^2^ + TMZ ^3^	II	Completed	[12]
NCT04277221	aDCs/tumour antigen + RT + TMZ	III	Unknown	[13]
NCT01213407	Trivax + RT + TMZ	II	Completed	[14]
NCT02772094	aDCs/tumour antigen + TMZ	II	Unknown	[15]
NCT01280552	ICT-107	II	Completed	[16]
NCT01006044	aDCs	II	Completed	[17]
NCT03382977	VBI-1901	I/II	Active, not recruiting	[18]
NCT02864368	PEP-CMV + TMZ	I	Terminated	
NCT02146066	DCVax-L + TMZ	E.A. ^4^	Available	
NCT04968366	aDCs pulsed with multiple neoantigen peptides	I	Recruiting	
NCT02709616	mRNA-pulsed DCs + RT + TMZ	I	Completed	[19]
NCT02808364	mRNA-pulsed aDCs	I	Completed	[19]
NCT02649582	mRNA-pulsed DCs + RT + TMZ	I/II	Recruiting	[20]
NCT02078648	SL-701	I/II	Completed	[21]
NCT05685004	TVI-Brain-1 + RT + TMZ	II/III	Not yet recruiting	
NCT02510950	Personalised peptide + Poly-ICLC + TMZ	I	Terminated	
NCT02465268	pp65-shLAMP DC	II	Active, not recruiting	[22]
NCT04801147	aDCs	I/II	Recruiting	
NCT01902771	DCs + tumour lysate + Imiquimod	I	Terminated	
NCT04002804	aDCs + autologous tumour lysate	I	Terminated	
NCT03665545	IMA950	I/II	Active, not recruiting	[23]
NCT04842513	Multipeptide plus XS15 + RT + TMZ	I	Recruiting	
NCT01290692	TVI-Brain-1	II	Completed	[24]
NCT02366728	CMV pp65 -LAMP mRNA-pulsed aDCs + TMZ + Basiliximab	II	Completed	[25]
NCT04116658	EO2401 + Nivolumab/Nivolumab + Bevacizumab	Ib/IIa	Active, not recruiting	[26]
NCT03916757	V-Boost	II	Unknown	[27]
NCT01567202	aDCs + Autogeneic glioma stem-like cells (A2B5+) + RT + TMZ	II	Unknown	[28]
NCT02010606	Allogenic GBM stem-like lysate-pulsed aDCs	I	Completed	[29]
NCT00643097	PEP-3-KLH + GM-CSF + TMZ	II	Completed	
NCT04963413	pp65-fLAMP RNA-loaded aDCs + GM-CSF + TMZ	I	Active, not recruiting	
NCT01957956	Allogenic tumour lysate-pulsed aDCs	Early I	Active, not recruiting	[30]
NCT01808820	aDCs + Allogenic tumour lysate + Imiquimod	I	Completed	
NCT04015700	GNOS-PV01 + INO-9012	I	Active, not recruiting	
NCT00639639	pp65-LAMP mRNA-loaded DCs + Ttd ^5^	I	Completed	[31]
NCT03360708	Allogenic tumour lysate-pulsed aDCs	Early I	Active, not recruiting	
NCT01081223	TVI-Brain-1 + IL-2	I/II	Completed	[24]
NCT02722512	HSPPC-96 + RT	I	Terminated	
NCT02455557	SurVaxM	II	Active, not recruiting	[32]
NCT03914768	Genetically modified tumour cells/antigens-pulsed aDCS	I	Unknown	
NCT03615404	aDCs + CMV RNA + CM-CSF + Ttd	I	Recruiting	
NCT02149225	APVAC + Poly-ICLC + GM-CSF	I	Completed	[33]
NCT03223103	Personalised vaccine + Poly-ICLC	I	Active, not recruiting	[34]
NCT05743595	Personalised neoantigen DNA + Retifanlimab	I	Not yet recruiting	
NCT03688178	aDCs + CMV pp65 + TMZ + Varlilumab	II	Recruiting	
NCT04888611	GSC-DCV + Camrelizumab	II	Recruiting	
NCT00890032	aDCs + Autologous tumour mRNA	I	Completed	
NCT00589875	AdV-tk + Valacyclovir	II	Completed	[35]
NCT01403285	IMA950 + Cyclophosphamide + GM-CSF	I	Terminated	
NCT03927222	aDCs + pp65-LAMP CMV mRNA + GM-CSF + Ttd	II	Terminated	
NCT00576537	aDCs + Allogenic tumour lysate	II	Completed	[36]
NCT04573140	RNA-LP	I	Recruiting	
NCT00751270	GliAtak	I	Completed	[37]
NCT03422094	NeoVax + Nivolumab + Ipilimumab	I	Terminated	
NCT04642937	GBM6-AD + Hp1a8 + Imiquimod	I	Active, not recruiting	[38]
NCT02052648	Indoximod + TMZ	I/II	Completed	[39]
NCT01222221	IMA950 + TMZ + RT	I	Completed	
NCT02529072	DCs + Nivolumab	I	Completed	
NCT00846456	CSC-mRNA transfected DCs	I	Completed	[40]
NCT01814813	MDNA-55	II	Terminated	[41]
NCT02287428	NeoVax + Pembrolizumab	I	Recruiting	
NCT03018288	HSPCC-96 + Pembrolizumab + RT + TMZ	I	Completed	
NCT04201873	aDCs + Pembrolizumab + Poly-ICLC	I	Recruiting	
NCT04280848	UCPvax	II	Recruiting	[42]
NCT01967758	ADU-623	I	Completed	[43]
NCT02820584	aDCs + GSC	I	Completed	
NCT04552886	aDCs	I	Recruiting	
NCT02718443	VXM01	I	Completed	[44]
NCT04523688	aDCs	II	Recruiting	
NCT01759810	DCs	I	Unknown	
NCT03750071	VXM01 + Avelumab	I/II	Active, not recruiting	[45]
NCT03879512	aDCs + Tumour lysate + Cyclophosphamide + Nivolumab + Ipilimumab	I/II	Recruiting	
NCT03395587	aDCs + SoC ^6^	II	Recruiting	
NCT00045968	DCVax-L	III	Active, not recruiting	[46]
NCT01498328	CDX-110 + Bevacizumab	II	Completed	[47]
NCT05698199	ITI-1001	I	Not yet recruiting	
NCT03149003	DSP-7888 + Bevacizumab	III	Completed	[48]
NCT00905060	HSPPC-96 + TMZ + Surgery	II	Completed	[49]
NCT00003185	Autologous tumour cells + Sargramostim	II	Completed	[50]
NCT00626015	PEP3-KLH + Daclizumab + TMZ	I	Completed	[51]
NCT00626483	DCs loaded with CMV pp65-LAMP mRNA	I	Completed	[52]
NCT00458601	CDX-110 + TMZ + GM-CSF	II	Complete	[53]
NCT05100641	AV-GBM-1	III	Not recruiting yet	
NCT01250470	SurVaxM	I	Complete	[54]
NCT03400917	AV-GBM-1	II	Active, not recruiting	[55]
NCT05163080	SurVaxM + Montanide + Sargramostim	II	Recruiting	
NCT01480479	CDX-110 + TMZ + GM-CSF	III	Completed	[56]
NCT00576641	aDCs + Autologous tumour lysate	I	Completed	[36]
NCT01204684	Resiquimod + Poly-ICLC	II	Active, not recruiting	
NCT01491893	PVSRIPO	I	Completed	[57]
NCT05557240	NPVAC1/2 + Poly-ICLC	I	Recruiting	
NCT04388033	aDCs + IL-12	I/II	Recruiting	
NCT05356312	Personalised neoantigen vaccine	E.A. ^4^	Available	
NCT05283109	P30-EPS + Hiltonol	I	Not yet recruiting	
NCT02498665	DSP-7888	I	Completed	[58]
NCT02800486	Cetuximab	II	Recruiting	
NCT04978727	SurVaxM	I	Recruiting	
NCT01920191	IMA950 + Poly-ICLC + TMZ	I/II	Completed	[59]
NCT00612001	aDCs	I	Completed	[60]
NCT00068510	aDCs loaded with tumour lysate	I	Completed	[61]
NCT04214392	CAR T-Cells	I	Recruiting	
NCT00293423	GP96	I	Completed	[62]
NCT01522820	DEC-205/NY-ESO-1 Fusion Protein CDX-1401 + Sirolimus	I	Completed	
NCT04808245	H3K27M peptide + Imiquimod	I	Recruiting	
NCT00069940	Telomerase: 540–548 peptide + GM-CSF	I	Completed	
NCT03043391	PVSRIPO	Ib	Active, not recruiting	[63]
NCT00014573	Surgery + Paclitaxel + Cyclophosphamide + Filgrastim + Autologous tumour cells + Sargramostim + Cisplatin + Carmustine + IL-2 + Autologous bone marrow/PBMC transplantation	II	Completed	
NCT01621542	WT2725	I	Completed	[64]
NCT00004024	Autologous tumour cells + Muromonab-CD3 + GM-CSF + IL-2	II	Completed	

^1^ Autologous dendritic cells; ^2^ radiotherapy; ^3^ temozolomide; ^4^ early access; ^5^ tetanus toxoid; ^6^ standard of care.

**Table 2 biomedicines-11-02100-t002:** Trials assessing exercise, diet, and microbiome in GBM patients.

ClinicalTrials.gov ID	Intervention	Phase	Stage	Reference
NCT03390569	Exercise	N/A ^1^	Completed	
NCT05015543	Personal training programme	N/A	Recruiting	
NCT02129335	Impact of exercise on stress	N/A	Terminated	[70]
NCT05431348	Impact of stress and exercise on chemoradiation outcome	N/A	Recruiting	[71]
NCT05116137	Circuit-based resistance exercise	N/A	Enrolling by invitation	[72]
NCT03501134	NovoTTF ^2^ device	N/A	Completed	
NCT01865162	Ketogenic diet	I	Completed	[73]
NCT05708352	Ketogenic diet	II	Not yet recruiting	[74]
NCT02286167	Atkins-based diet	N/A	Completed	[75]
NCT02939378	Ketogenic diet	I/II	Unknown	
NCT03075514	Ketogenic diet	N/A	Completed	[76]
NCT02302235	Ketogenic diet	II	Completed	[73]
NCT00508456	Methionine-restricted diet	I	Terminated	
NCT04730869	Metabolic therapy programme	N/A	Recruiting	[77]
NCT00575146	Ketogenic diet	I	Completed	[78]
NCT04691960	Ketogenic diet + Metformin	II	Recruiting	
NCT01535911	Metabolic nutritional therapy	N/A	Active, not recruiting	[79]
NCT02046187	Ketogenic diet	I/II	Terminated	[80]
NCT03451799	Ketogenic diet	I	Active, not recruiting	[81]
NCT05183204	Ketogenic diet + Metformin + Paxalisib	II	Recruiting	
NCT03160599	Ketogenic diet	N/A	Unknown	
NCT03278249	Modified Atkins ketogenic diet	N/A	Active, not recruiting	
NCT02768389	Modified Atkins diet + Bevacizumab	Early I	Completed	
NCT01754350	Calorie-restricted ketogenic diet + Transient fasting	N/A	Completed	[82]
NCT00243022	*Boswellia serrata* extract + Vitamin B12	II	Terminated	[83]
NCT05326334	Chemoradiation + Chemotherapy + Microbiome evaluation	N/A	Recruiting	
NCT00003751	Penicillamine + Low copper diet	II	Completed	[84]
NCT03631823	Chemotherapy and/or radiotherapy + Correlation between microbiome and prognosis	N/A	Unknown	

^1^ Not applicable; ^2^ tumour treating fields.

**Table 3 biomedicines-11-02100-t003:** Immunotherapy trials for TNBC.

ClinicalTrials.gov ID	Vaccine	Phase	Stage	Reference
NCT04674306	α-lactalbumin + Zymosan	Early I	Recruiting	[97]
NCT04024800	AE37 peptide + Pembrolizumab	II	Active, not recruiting	[98]
NCT03199040	Neoantigen DNA + Durvalumab	I	Active, not recruiting	
NCT04348747	HER2/HER3 DCs + Pembrolizumab	IIa	Recruiting	[99]
NCT02348320	Polyepitope DNA	I	Completed	
NCT02938442	P10s-PADRE with MONTANIDE ISA 51 VG + Doxorubicin + Cyclophosphamide + Paclitaxel + Surgery	II	Completed	[100]
NCT03362060	PVX-410 + Pembrolizumab	Ib	Active, not recruiting	[101]
NCT02826434	PVX-410 + Durvalumab + Poly-ICLC	Ib	Active, not recruiting	[102]
NCT05455658	STEMVAC + Sargramostim	II	Recruiting	[103]
NCT03606967	Personalised neoantigen peptide + Carboplatin + Gemcitabine + Nab-Paclitaxel + Durvalumab + Tremelimumab + Poly-ICLC	II	Recruiting	
NCT03387085	N-803 + ETBX-011 + ETBX-051 + ETBX-061 + GI-4000 + GI-6207 + GI-6301 + HaNK + Avelumab + Bevacizumab + Aldoxorubicin + Capecitabine + Cisplatin + Cyclophosphamide + 5-Fluorouracil + Leucovorin + Nab-Paclitaxel	Ib/II	Active, not recruiting	
NCT03012100	Multi-epitope folate receptor alpha + Cyclophosphamide + GM-CSF	II	Active, not recruiting	[104]
NCT00986609	MUC1 + Poly-ICLC	Early I	Completed	[105]
NCT02593227	Folate receptor alpha + Cyclophosphamide + GM-CSF	II	Completed	[104]
NCT05504707	HER2-/HER3-primed DC1	I	Recruiting	
NCT04105582	Neoantigen-pulsed aDCs ^1^	I	Completed	
NCT02018458	Cyclin B1/WT1/CEF-pulsed DCs + Doxorubicin + Cyclophosphamide + Paclitaxel + Carboplatin	I	Completed	[106]
NCT02316457	RNA for shared tumour associated antigens + RNA for tumour specific antigens	I	Active, not recruiting	[107]
NCT03562637	OBI-822 + OBI-821	III	Recruiting	[108]
NCT04634747	PVX-410 + Pembrolizumab + Chemotherapy	II	Not yet recruiting	
NCT05269381	Personalised neoantigen + Pembrolizumab + Cyclophosphamide + GM-CSF	I	Recruiting	
NCT03761914	Galinpepimut-S + Pembrolizumab	I/II	Active, not recruiting	[109]
NCT02432963	P53MVA + Pembrolizumab	I	Active, not recruiting	[110]
NCT05329532	Modi-1/Modi-1v + Pembrolizumab	I/II	Recruiting	[111]
NCT00640861	MUC1 + HER2/neu + CpG + GM-CSF + IFA	Early I	Completed	[112]
NCT04879888	Peptide-pulsed aDCs	I	Completed	
NCT05035407	KK-LC-1 TCR + Aldesleukin + Cyclophosphamide + Fludarabine	I	Recruiting	

1 Autologous dendritic cells.

**Table 4 biomedicines-11-02100-t004:** Trials assessing exercise, diet, and microbiome in TNBC patients.

ClinicalTrials.gov ID	Intervention	Phase	Stage	Reference
NCT01498536	Aerobic exercise	N/A ^1^	Completed	[147]
NCT03733119	Methionine-restricted diet + ONC201	II	Terminated	[148]
NCT04248998	Fasting-mimicking diet + Metformin	II	Active, not recruiting	[149]
NCT05763992	Fasting-like approach + SoC ^2^	II	Recruiting	
NCT03186937	Methionine-restricted diet	II	Terminated	
NCT02348320	Caloric restriction diet + SABR ^3^	II	Recruiting	
NCT04677816	Vitamin D_3_ + SoC	II	Recruiting	
NCT05198843	Icosapent ethyl + Dasatinib	Ib/II	Recruiting	
NCT05037825	ICI ^4^ + Microbiome evaluation	N/A	Recruiting	
NCT03586297	SoC + Correlation between microbiome composition and pCR ^5^	N/A	Recruiting	
NCT04638751	Chemotherapy + Correlation between microbiome, PFS, ^6^ and OS ^7^	N/A	Recruiting	
NCT05916755	Pembrolizumab and/or chemotherapy + microbiome analysis to establish predictive biomarkers	N/A	Recruiting	
NCT03289819	Pembrolizumab + Nab-Paclitaxel + Epirubicin + Cyclophosphamide + Correlation between microbiome and clinical outcome	II	Completed	[150]

^1^ Not applicable; ^2^ standard of care; ^3^ stereotactic ablative radiotherapy; ^4^ immune checkpoint inhibitors; ^5^ pathological complete response; ^6^ progression-free survival; ^7^ overall survival.

**Table 5 biomedicines-11-02100-t005:** Immunotherapy trials for PCa.

ClinicalTrials.gov ID	Vaccine	Phase	Stage	Reference
NCT00003871	Fowlpox prostate specific antigen	II	Completed	[156]
NCT00374049	MUC1 + Poly-ICLC + GM-CSF	I	Completed	
NCT00122005	GVAX	I/II	Unknown	
NCT03815942	ChAdOx1-MVA 5T4 + Nivolumab	I/II	Unknown	[157]
NCT02234921	Cyclophosphamide + Dribble + Imiquimod + Cervarix	I	Completed	
NCT01867333	Enzalutamide + PROSTVAC-F/TRICOM + PROSTVAC-V/TRICOM	II	Completed	[158]
NCT04914195	Leuprolide acetate	III	Recruiting	
NCT01420965	Sipuleucel-T + CT-011 + Cyclophosphamide	II	Terminated	
NCT00292045	NY-ESO-1 protein + CpG 7909	I	Completed	[159]
NCT00140348	GVAX	I/II	Completed	
NCT00140400	GVAX	I/II	Completed	
NCT01095848	DPX-0907	I	Completed	[160]
NCT00089856	GVAX	III	Terminated	[161]
NCT00133224	GVAX	III	Terminated	[162]
NCT00005039	Fowlpox prostate specific antigen	II	Terminated	
NCT00906243	CV9103	I/II	Terminated	[163]
NCT05104515	OVM-200	I	Recruiting	
NCT03384316	ETBX-051 + ETBX-061 + ETBX-011	I	Completed	[164]
NCT03338790	Nivolumab + Rucaparib Nivolumab + Docetaxel + Prednisone Novilumab + Enzalutamide	II	Active, not recruiting	[156,157]
NCT03879122	ADT + Docetaxel ADT + Docetaxel + Nivolumab ADT + Ipilimumab/Docetaxel + Nivolumab	II/III	Active, not recruiting	
NCT04382898	BNT112 +/− Cemiplimab	I/II	Recruiting	[158]
NCT04077021	CCW702	I	Terminated	[159]
NCT03805594	[^177^Lu]-PSMA-617 + Pembrolizumab	Ib	Active, not recruiting	[160]
NCT04100018	Nivolumab + Docetaxel + Prednisone	III	Recruiting	
NCT03637543	Nivolumab	II	Recruiting	[161]
NCT05580107	MDPK67b	I	Recruiting	

**Table 6 biomedicines-11-02100-t006:** Trials assessing exercise, diet, and microbiome in PCa patients.

ClinicalTrials.gov ID	Intervention	Phase	Stage	Reference
NCT03658486	Exercise	N/A ^1^	Recruiting	
NCT03880422	Aerobic and resistance exercise + diet	N/A	Recruiting	
NCT02233608	Advanced pelvic floor muscle exercise	I/II	Completed	[168]
NCT01973673	Bone health educational materials	N/A	Completed	[169]
NCT05612880	Physical function assessment following androgen receptor signalling inhibitors	N/A	Recruiting	
NCT03397030 *	Exercise	N/A	Completed	[170]
NCT00660686 *	Resistance exercise + Flexibility training	N/A	Completed	[171]
NCT01696539	SoC ^2^ + Walking intervention	N/A	Completed	[172]
NCT00658229	Strength training group	III	Completed	[173]
NCT00253916 *	Aerobic cardiovascular exercise Resistance exercise	N/A	Completed	[174]
NCT02453139	Aerobic exercise	N/A	Completed	[175]
NCT00329797	Zoledronic acid and/or Calcium + Vitamin D	III	Completed	
NCT02710721	Fasting	N/A	Completed	
NCT02946996	Metformin + oligomeric procyanidin complex	II	Recruiting	
NCT03709485 *	Correlation between microbiota and development of prostate cancer	N/A	Unknown	[176]
NCT04687709 *	Correlation between microbiome and ADT-related metabolic changes	N/A	Recruiting	[177]
NCT04638049 *	Correlation between microbiota/metabolome and radiation-induced gastrointestinal toxicities	N/A	Completed	

^1^ Not applicable; ^2^ standard of care; * not specified as advanced prostate cancer.

## Data Availability

Not applicable.

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
