# Peer review of "A Holistic Approach to Hard-to-Treat Cancers: The Future of Immunotherapy for Glioblastoma, Triple Negative Breast Cancer, and Advanced Prostate Cancer"

_biomedicines, 2023, doi:10.3390/biomedicines11082100_

Round 1
Reviewer 1 Report
The manuscript describes the relationship of extrinsic factors in glioblastoma, TNBC, and advanced prostate cancer. The authors establish that these types of cancer have an improvement when exercise and adjuvant microbiome treatment are included. The authors present three tables where the different clinical trials carried out for glioblastoma, TNBC and prostate cancer are indicated.
However, the authors point to very few studies involving the action of exercise and the modulation of the microbiome in these types of cancer.
1. In order to strengthen the manuscript, the authors must include three tables that indicate those clinical trials that incorporate exercise or microbiome modulation in glioblastoma, TNBC, and prostate cancer.
For example, there are the following clinical trials for glioblastoma, exercise, diet, or gut microbiota: NCT03390569, NCT05116137, NCT04730869, NCT05708352, NCT05015543, NCT02129335, NCT03501134, NCT05326334, NCT0363182 3 etc
For TNBC there are some clinical trials that address the issue in question: NCT03586297, NCT05037825, NCT01498536, etc.
In prostate cancer, the list of clinical trials addressing the subject is broader: NCT03397030, NCT00660686, NCT01696539, NCT00658229, NCT00253916, NCT02453139, NCT03709485, NCT04687709, NCT04638049, etc.
Author Response
We thank the reviewer for their time in reviewing this manuscript. We have followed the reviewer’s advice and added 3 tables (one for each difficult to treat type of cancer, i.e GBM, TNBC and advanced prostate cancer) highlighting clinical trials currently assessing exercise, diet and microbiome. Many of these approaches have been examined as potential therapies on their own as opposed to be combined with antigen specific immunotherapies or used to “prepare” the patients prior to the patients receiving active immunotherapeutic treatment.
Reviewer 2 Report
* This topic is very interesting and well organized. The reason for choosing triple-negative breast cancer (TNBC), glioblastoma multiforme (GBM), and advanced prostate cancer (PCa). How about other types of breast cancer?
* I suggest making a figure revealing every mechanism of this review, especially in the section on Chronic Stress and Cancer.
* The title of this manuscript is The Future of Immunotherapy so, I suggest adding a separate section about the future perspectives in the immunotherapy of breast cancer.
* Please, use American terms instead of British terms such as using tumor instead of tumour.
Author Response
We thank the reviewer for their time and effort in reviewing this manuscript We have prioritised and clearly stated in the abstract that the review will focus solely on TNBC as a hard-to-treat type of breast cancer.
We have now added a figure summarising the main mechanisms behind the positive and negative modulation of the immune system driven by life-style factors which could be harnessed to better “prepare” patients before they receive any form of active vaccination.
We have followed the guidelines provided by the journal which stipulate that either British or American English spelling can be used as long as the entire manuscript is consistent.
Round 2
Reviewer 2 Report
* The manuscript has been modified greatly. I have a last minor issue with the title. The term "Hard-to-treat" is too broad including many diseases. The authors must specify the three diseases included in this review only.
Author Response
Dear Reviewer,
We acknowledge that the review focused on TNBC, GBM and advanced prostate cancer and have changed the title accordingly.
Many thanks,
Stephanie.